# Essential Oils as Antimicrobial Active Substances in Wound Dressings

**DOI:** 10.3390/ma15196923

**Published:** 2022-10-06

**Authors:** Daniela Gheorghita, Elena Grosu, Alina Robu, Lia Mara Ditu, Iuliana Mihaela Deleanu, Gratiela Gradisteanu Pircalabioru, Anca-Daniela Raiciu, Ana-Iulia Bita, Aurora Antoniac, Vasile Iulian Antoniac

**Affiliations:** 1Faculty of Material Science and Engineering, University Politehnica of Bucharest, 313 Splaiul Independentei Street, 060042 Bucharest, Romania; 2Faculty of Biology, University of Bucharest, 1-3 Intr. Portocalelor Street, 060101 Bucharest, Romania; 3Faculty of Applied Chemistry and Materials Science, University Politehnica of Bucharest, 1-7 Polizu Street, 011061 Bucharest, Romania; 4Research Institute of the University of Bucharest, 90 Sos. Panduri, 050663 Bucharest, Romania; 5Academy of Romanian Scientists, 54 Splaiul Independentei Street, 050094 Bucharest, Romania; 6Faculty of Pharmacy, Titu Maiorescu University, 22 Dambovnicului Street, 040441 Bucharest, Romania; 7S.C. Hofigal Import Export S.A., 2 Intrarea Serelor Street, 042124 Bucharest, Romania

**Keywords:** wound healing, hydrogels, biopolymers, essential oils, microencapsulation, antimicrobial properties, cytotoxicity, MTT assay

## Abstract

Wound dressings for skin lesions, such as bedsores or pressure ulcers, are widely used for many patients, both during hospitalization and in subsequent treatment at home. To improve the treatment and shorten the healing time and, therefore, the cost, numerous types of wound dressings have been developed by manufacturers. Considering certain inconveniences related to the intolerance of some patients to antibiotics and the antimicrobial, antioxidant, and curative properties of certain essential oils, we conducted research by incorporating these oils, based on polyvinyl alcohol/ polyvinyl pyrrolidone (PVA/PVP) biopolymers, into dressings. The objective of this study was to study the potential of a polymeric matrix for wound healing, with polyvinyl alcohol as the main material and polyvinyl pyrrolidone and hydroxypropyl methylcellulose (HPMC) as secondary materials, together with additives (plasticizers poly(ethylene glycol) (PEG) and glycerol), stabilizers (Zn stearate), antioxidants (vitamin A and vitamin E), and four types of essential oils (fennel, peppermint, pine, and thyme essential oils). For all the studied samples, the combining compatibility, antimicrobial, and cytotoxicity properties were investigated. The obtained results demonstrated a uniform morphology for almost all the samples and adequate barrier properties for contact with suppurating wounds. The results show that the obtained samples containing essential oils have a good inhibitory effect on, or antimicrobial properties against, *Staphylococcus aureus ATCC 25923, Enterococcus faecalis ATCC 29212, Escherichia coli ATCC 25922, Pseudomonas aeruginosa ATCC 27853,* and *Candida albicans ATCC 10231.* The MTT assay showed that the tested samples were not toxic and did not lead to cell death. The results showed that the essential oils used provide an effective solution as active substances in wound dressings.

## 1. Introduction

Nowadays, due to stress and inadequate nutrition, the health of a considerable number of people is severely affected by cardiovascular disease and diabetes, which in their advanced stages can lead to unmanageable complications. Additionally, the problem of infections that can appear after a surgical intervention is important in all medical fields, such as orthopedics [1,2,3,4,5,6], dentistry [7,8], neurosurgery [9,10], or general surgery [11,12,13,14,15]. Ulcers in the case of stroke or pressure ulcers due to diabetes constitute the third most expensive disease. Patients experience reduced physical activity, reduced ability to eat, urinary and fecal incontinence, and decreased consciousness. Last but not least, ulcers and pressure ulcers lead to almost unbearable pain, especially for immobilized patients. Pressure ulcers or ulcers are skin lesions that can be classified into types 1, 2, 3, and 4 depending on the depth of the tissue lesion. Type 1 is the least severe stage, and type 4 is defined as the complete destruction of the tissue [16,17].

It is estimated that, annually, the rate of mortality due to the complications of these diseases has increased 2–6 times more than that due to other diseases. For these reasons, it is important to manage bedsores and pressure ulcers in order to estimate the duration of wound closure, as many wounds do not close during hospitalization. Therefore, by controlling the damaged area of the skin, objective and reproducible information can be obtained, which can provide a healing prognosis [18,19,20,21,22,23,24,25].

The market for medical products used for treating bedsores and pressure ulcers is dominated by dressings, which must meet several requirements, such as the ability to absorb exudate from the wound to maintain a moist but not macerated wound, as well as water permeability, pain relief, and wound pH optimization. The efficiency of the dressings depends on the key material and the additives used as their components [26]. Currently, the possibility of using phytochemicals in the form of plant extracts for the treatment of open skin wounds is gaining increasing therapeutic importance. In particular, the essential oils (EO) extracted from different parts of plants contain many active compounds with antioxidant and antimicrobial properties. Numerous studies have aimed to replace antibiotics and other synthetic compounds in the treatment of microbial infections [27]. At the same time, special importance is given to the selection of the materials for preparing the polymer matrix, as they play a significant role in the treatment of skin lesions and are also a suitable vehicle for the administration of essential oils directly to the healing site [28].

Among the existing dressings on the market are hydrocolloid dressings, which contain gel for the development of new skin cells in ulcers; alginate dressings, which accelerate the healing process; and dressings containing silver nanoparticles with antibacterial properties, creams, and ointments to prevent tissue damage and accelerate the process of healing.

The most prevalent compounds, in regard to the functionalization of wound dressings, are antimicrobial agents. These agents provide antimicrobial properties to wound dressings and are divided into three groups: antibiotics (e.g., tetracycline, gentamicin), natural biological materials (e.g., essential oils, honey) and nanoparticles (e.g., silver, gold) [29,30]. Natural and synthetic biomaterials are two main categories of biomaterials used for wound dressings. The most common natural biomaterials used for wound dressings are collagen, hyaluronic acid, chitin, chitosan, starch, gelatin, and alginate. These types of dressings are better in terms of their biocompatibility, antibacterial activity, antioxidation, hemostasis, and capacity for healing promotion [31]. Caridade et al. [32] developed thick free-standing membranes formed of alginate and chitosan multilayer films. They concluded that these membranes are biocompatible and highly stable when used in a physiological buffer, offering new potential for wound healing and tissue engineering applications. However, synthetic polymer-based dressings can provide a broader spectrum of mechanical properties compared to natural wound dressings. Polylactic acid (PLA), polycaprolactone (PCL), and polyethylene glycol (PEG) are examples of synthetic polymers that have been widely studied for wound dressing applications [33,34,35]. In a study developed by Bardania et al. [36], the novel strategy of using synthesized silver nanoparticles (AgNPs) incorporated into PLA/PEG nanofilm showed promising outcomes. The biocompatible silver nanoparticles were synthesized using Teucrium polium extract as a reducing agent, an approach that proved to be efficient and cost-effective. The nanofilm displayed promising antimicrobial and antioxidant properties, showing strong potential as a wound dressing.

Both natural and synthetic biomaterials possess advantages and disadvantages, and this is why research is now focusing on combining different types of polymers in order to improve their wound healing properties and control their biodegradation and drug release [37].

Amalraj et al. developed biocomposite films by incorporating black pepper essential oil and ginger essential oil into polyvinyl alcohol (PVA), gum arabic (GA), and chitosan (CS). Obtained by the solvent casting method, the biocomposite films showed enhanced mechanical properties with an improved heat stability, as well as antibacterial activity against Gram-positive and Gram-negative bacteria [38].

At present, patients with open skin lesions who have been hospitalized with a diagnosis of stroke or diabetes are exposed to an increasing number of microbial strains that are resistant to antibiotics and main pathogenic agents of healthcare-associated infections. For these patients, healing involves the management of bedsores or pressure ulcers, along with drug treatment for the disease itself. Healing may be delayed in patients with hypersensitivity to antibiotics. Many cases of resistance to vancomycin in Gram-positive bacteria (*Staphylococcus aureus*) and resistance to ciprofloxacin in Gram-negative bacteria (*Pseudomonas aeruginosa*) have been reported. To solve these inconveniences and to shorten the healing time of wounds, we are currently working on developing dressings containing active essential oils with antimicrobial components. The modernization of the processes of harvesting and processing medicinal and aromatic plants has led to the preservation of their antioxidant and nutritional properties and, therefore, to an increase in the number of possibilities to capitalize on these substances in numerous pharmaceutical and cosmetic fields. At the same time, due to the growing interest in improving environmental conditions, we are assisting in the development of green pharmacy, in which chemical compounds are replaced by phytopharmaceuticals [39,40].

The production of phytopharmaceutical, cosmetic, and nutraceutical preparations has led to an annual increase in turnover, as the population opts for natural products due to numerous advantages, such as their rare adverse effects, non-addictive quality, wide accessibility, possibility of being used as a treatment in combination with other therapies and diets, and lower prices [41,42]. Worldwide, the market for the consumption of herbal products is constantly growing, which is why the cultivation of medicinal plants can provide a good business opportunity [43].

The products extracted from medicinal and aromatic plants have different uses depending on the type and quantity of active principles contained. They vary depending on the biological conditions of the plant species and the cultivation area due to the soil quality and climatic conditions. The optimization of the technological flow of the processing of harvested products and their conditioning and storage is of great importance [44].

Crops such as vegetables, fruits, cereals, and vines undergo harvest in maturity. Crops contain many natural components, such as liquid, oil, fatty acids, amino acids, organic acids, various cyclic ethers, glycerol, vitamins (B, C, and E) [45], aldehydes and ketones (esters and ethers), hydrocarbons, aromatics, vegetal proteins, polysaccharides, lignin, minerals, isoflavones, polyols, alcohols (aromatic or terpenic), carbohydrates, amino acids (myristic, palmytic, palmitoleic, heptadecanoic, stearic, oleic, linoleic, linolenic, arachidic, eicosenoic, behenic) [46], open and closed chain terpenes, sesquiterpenes, nitrogen, and sulphate. Among the many constituents of essential oils, we can list limonene, α-pinene, geraniol, menthol, eucalyptol, and thymol [47]. The medical applications of some essential oils are presented in Table 1.

The incorporation of essential oils into polymeric compositions can be performed by blending them with components so as to ensure an at least 10 wt. % content in the matrix [49]. A modern embedding method is to encapsulate essential oils in a polymeric shell, such as sodium alginate or cyclodextrins [50,51,52,53]. With respect to polymeric materials used in matrices for wound dressing applications, the most frequently used are polyvinyl alcohol, polyvinyl pyrrolidone [54,55,56,57,58], chitosan [59,60,61], alginate [62,63,64,65], cyclodextrines [66,67,68,69,70], hydroxypropyl methyl cellulose (HPMC) [71,72,73], polycaprolactones [74,75,76], poloxamer [77,78,79,80]. In order to ensure flexibility of wound dressing, polymeric compositions are mixed with plasticizers type polyethylene glycol (PEG) [81], glycerol [82], citrates [83], and adipates [84].

Blends of polyvinyl alcohol (PVA) and polyvinyl pyrrolidone (PVP), as highly hydrophilic and biocompatible polymers, are well known in the literature [85] and are used as matrix network structures to obtain films or hydrogels with improved mechanical properties for medical applications as drug delivery [55]. Plasticizer PEG and glycerol are biocompatible and are often used in polymers to confer flexibility on the material [86]. Due to its low molecular mass, PEG decreases the glass transition temperature in polymers. Complexes of PEG and glycerol can improve certain properties of polymers, such as their elongation at break, ability to form a film, flexibility, and softness [87,88,89,90,91]. HPMC is a non-ionic polymer that can be used as a thickening agent, film additive, and stabilizer [72]. Vitamin A and vitamin E are well known for their antioxidant properties and wide applicability in cosmetics and drug delivery [92,93]. Zn stearate also is a common stabilizer used in the formulation of polymeric compositions with medical applications [94,95].

The objective of this study was to develop select polymeric matrices for wound healing, with the main material being polyvinyl alcohol and the secondary materials being polyvinyl pyrrolidone and hydroxypropyl methyl cellulose (HPMC), together with additives, including plasticizers poly(ethylene glycol) (PEG) and glycerol, stabilizers (Zn stearate), antioxidants (vitamin A and vitamin E), and four types of essential oils, in order to evaluate their combining compatibility and antimicrobial and cytotoxicity properties. In order to prepare antimicrobial wound dressings, we selected peppermint, thyme, pine, and fennel essential oils as antimicrobial active substances to be loaded into polymer matrices. We made films from a control sample containing PVA/PVP and additives, four film samples starting from the control sample, in which we introduced fennel, pine, mint and thyme essential oils by mixing with the control sample and three film samples in.

We created films from a blank sample containing PVA/PVP and additives, obtaining four film samples starting with the blank sample, into which we introduced essential oils of fennel, pine, mint, and thyme by mixing them with the blank sample, and three film samples into which we introduced the pine, peppermint, and thyme essential oils encapsulated in sodium alginate. An important stage of the development of the materials was the encapsulation of the essential oils. The obtained microcapsules were embedded in the polymer matrix for a prolonged release of essential oils, including pine, mint, and thyme encapsulated in sodium alginate.

We performed physical-chemical and biocompatibility analyses of samples of the prepared compositions in order to estimate the performance of the essential oils as antimicrobial active substances. The essential oils were uniformly distributed in the polymeric matrices both in liquid form and encapsulated in sodium alginate.

## 2. Materials and Methods

In order to prepare the polymeric compositions to be used for the healing of skin lesions, we used the following medical-grade materials: polyvinyl alcohol (PVA) (US Pharmacopeia, Sigma-Aldrich, St. Louis., MO, USA), average Mw 85,000–124,000, 99.8% hydrolyzed, relative density 1.269 g/cm^3^); polyvinyl pyrrolidone (PVP) (K90, EM powder, Mw = 360,000, Sigma-Aldrich); (hydroxypropyl)methyl cellulose (Sigma-Aldrich); sodium alginate (Sigma-Aldrich); calcium chloride (Sigma-Aldrich); ethanol absolute (99.8%, Honeywell, Charlotte, NC, USA); Tween 80 viscous liquid (Sigma-Aldrich); glycerol (Honeywell); polyethylene glycol (PEG) average Mn 400 (Sigma-Aldrich); glutaraldehyde (GTA) solution grade II, 25% in H_2_O (Sigma-Aldrich); vitamin A dermal application (Biogalenica); vitamin E (TRIOVERDE); essential oils (fennel obtained from *Foeniculum vulgare*, peppermint obtained from *Mentha piperita*, pine obtained from *Pinus sylvestris*, and thyme obtained from *Thymus* sp.); and deionized water produced in a laboratory. The essential oils were manufactured by S.C. HOFIGAL S.A. Bucharest using the technology of extraction with solvents from the aerial parts of plants. The chemical formulas of the essential oils used here are presented in Figure 1.

The characteristics and compositions of the essential oils, determined in the HOFIGAL laboratories, are presented in Table 2 and Table 3.

***Preparing of the samples.*** A blend of PVA/PVP (60:40) was dissolved in deionized water at (temperature) 90 °C under stirring at 280 rpm to prepare the initial control sample, E0_1. Through Fourier-transform infrared spectroscopy (FTIR), a polymeric film was obtained from E0_1 by the casting method. Furthermore, the composition E0_2 was obtained after mixing the E0_1 composition at 35 °C for 3 h with glutaraldehyde (GTA) [104]. A film that was 164 µm thick was prepared by the casting method to be used for the FTIR investigations. The film of E0_2 was cut in order to be weighed and immersed in distilled water for one hour. The sample resulting from the immersion in distilled water was dried and weighed again to determine the degree of crosslinking. E1 is the leading sample. 

Two types of compositions containing essential oils were prepared, starting with the initial composition, E1, as follows: compositions E2, E4, E6, and E8, to which essential oils were added, and the compositions E3, E5, and E7, into which were introduced microcapsules loaded with essential oils (Table 4). 

The microcapsules acquired from sodium alginate were loaded with essential oils. The emulsion comprised of 50:50 essential oil/ethanol and the surfactant additive Tween 80 to stabilize the emulsion, and it was mixed with the mixer IKA T18 Digital Ultra Turax at 23,000 rpm for half an hour at room temperature. The crosslinking agent was prepared from calcium chloride dissolved in deionized water [105]. Microcapsules filled with peppermint, pine, or thyme essential oils were formed by dripping the mixture of sodium alginate solution and essential oil emulsion into a calcium chloride solution (Figure 2). At the end of the procedure, the obtained capsules were washed with distilled water by pouring. The capsules obtained were added to the polymeric composition through gentle mixing. The films of compositions E3, E5, or E7 were obtained by the casting method. 

A flow chart of the procedure used to develop samples E1–E8 is presented in Figure 3.

### 2.1. Contact Angle

The contact angle measurements of the polymeric matrices were performed using the experimental KRÜSS DSA30 Drop Shape Analysis System. Since the polymeric matrices were designed to come into contact with biological fluids, the liquid used to determine the contact angles was distilled water. Distilled water drops of approximately 10 µL flowed from the dosing system of the apparatus and angle values were recorded for 30 s continuously at intervals of 2 s. The contact angle values were reported as averages.

### 2.2. Optical Microscopy

The characterization of the structure of the polymeric matrices, additives, and essential oil dispersion and the measurement of the size of the alginate microcapsules loaded with essential oils were performed using an optical microscope, Olympus BX51 (Olympus Life and Materials Science Europa GMBH, Hamburg, Germany), equipped with lenses for magnifications of 20×, 10× and 5×, and an Analysis S.1 from Olympus Soft Imaging Solutions GmbH.

### 2.3. Film Thickness

The thickness of the obtained films/membranes was measured with a micro-meter (Mitutoyo Mfg Co. Ltd., Kawasaki, Japan). The value of the thickness was calculated as the average of five measurements taken at different positions.

### 2.4. Gel Fraction

A sample from E0_2 was dried, weighed (*m_i_*), and immersed in deionized water for 24 h at room temperature and then weighed again (*m_f_*) (g). The gel fraction was calculated with the following formula [106]:(1)$$\mathrm{Gel}\;\mathrm{fraction}\;(\%)=\frac{m_{f}}{m_{i}}\times 100$$

### 2.5. Swelling Ability

In order to calculate the swelling degree of the films, each one was cut into 3 cm-diameter discs and developed to obtain a constant weight through drying. The dried pieces were again immersed in deionized water at room temperature (20 °C). The experiments were conducted in triplicate, and the measurements were performed until a constant mass of the swollen samples (2 to 5 h) was obtained. Since the time needed to reach the equilibrium was very short, the mass of the dissolved polymer was considered negligible. Additionally, to keep the calculation simple, we considered that the amount of active principle contained within the samples and released into the experimental aqueous environment was insignificant compared to the total amount of water that was absorbed. The swelling degree (SD) was determined with Equation (2), where *m_i_* (g) is the initial sample weight and *m_s_* (g) represents the swollen sample weight [107,108]:(2)$$SD=\frac{m_{s}-m_{i}}{m_{i}}\times 100$$

### 2.6. Water Solubility

Water solubility measurements were performed in triplicate, in accordance with the method of Shen et al., with some modifications, as detailed below [108]. The film samples (2 × 2 cm), having been previously dried to a constant weight, were introduced into conical flasks that contained the same measured amount of deionized water (100 mL). The flasks were kept at room temperature and continuously shaken for 24 h. The remaining undissolved pieces of film were removed from the water and again dried to a constant weight. The water solubility (WS) of the composite materials was calculated based on the weight of the initial dried sample, *m_i_* (g), and the weight of dried undissolved sample, *m_f_* (g), using Equation (3):(3)$$WS=\frac{m_{i}-m_{f}}{m_{i}}\times 100$$

### 2.7. Water Vapor Permeability 

For each film, the water vapor permeability (*WVP*) was calculated using Equation (4), following the method previously described by Limpan et al. [109]. The active films were cut to a diameter of 3 cm and sealed in polypropylene beakers containing silica gel (0% RH). The beakers were then placed in a desiccator containing distilled water at room temperature (20 °C) and in 95% relative humidity for six days. The desiccator chamber was equipped with temperature and relative humidity sensors. Periodically, the absorbed moisture was determined gravimetrically. The test was performed in triplicate [110]:(4)$$WVP=WVTR\times \frac{\delta }{\Delta p}$$
where *WVTR* is the water vapor permeability (g m^−2^ h^−1^), *δ* is the thickness of the film (m), and Δ*p* represents the water vapor partial pressure difference between the two sides of the coatings [Pa].

### 2.8. Fourier-Transform Infrared Spectroscopy Analyses (FTIR)

FTIR analyses were performed to characterize the internal structure of the functional groups in the initial control sample E0_1 (polymeric blend of PVA/PVP), as well as control sample E0_2, to highlight the crosslinking with the glutaraldehyde of the composition E0_1. Afterwards, the following FTIR analyses were performed in order to highlight the interaction between the initial matrix PVA/PVP with the additives and essential oils in compositions E1–E8. The FTIR absorption spectra were obtained using the spectrometer JASCO FTIR 6200 and were in the spectral range of 4000 cm^−1^–600 cm^−1^.

### 2.9. Scanning Electron Microscopy and Energy Dispersive X-ray Analysis 

A scanning electron microscope (SEM), the QUANTA INSPECT F Scanning Electron Microscope (FEI Company, Eindhoven, The Netherlands), equipped with an energy dispersive X-ray spectrometer detector (EDAX) (FEI Company, Eindhoven, The Netherlands) with a 132 eV resolution at MnKα, was used to obtain information about the inner morphology of the polymeric matrices and also to characterize the appearance of the surfaces. In order to perform the SEM analyses, the polymeric surfaces were covered with gold to lower the electronic charging effect.

### 2.10. Microbiological Analyses


*Antimicrobial efficiency assays*


The antimicrobial assays were performed using standard strains of *Staphylococcus aureus* ATCC 25923, *Enterococcus faecalis* ATCC 29212, *Escherichia coli* ATCC 25922, *Pseudomonas aeruginosa* ATCC 27853, and *Candida albicans* ATCC 10231. The microbial strains were selected from the collection of microbiology department (University of Bucharest), according to “SREN 14885—Chemical Disinfectants and Antiseptics—Application of European Standards for Chemical Disinfectants and Antiseptics” [111]. 

*The qualitative screening* was performed using an adapted spot diffusion method [112]. The microbial suspensions, with a standard density of 0.5 McFarland (corresponding to 1.5 × 10^8^ CFU/mL), were obtained from fresh 24–48 h bacterial and yeast cultures. The microbial inoculums were seeded on Petri dishes containing Muller Hinton agar (for the bacterial strains) [113,114,115] or Sabouraud agar (for the yeast strain) [116], according to the CLSI diffusion method (Clinical Laboratory Standard Institute, 2022) [117,118], and the sterile material specimens with a surface area of 1 cm^2^ were disposed on the surface of the inoculated medium and incubated for 20 h at 37 °C.

In order to evaluate the antimicrobial activity using the viable cell count (VCC) method [119], the materials were immersed in 1 mL of appropriated broth medium, which was subsequently inoculated with microbial suspension at a final density of 1.5 × 10^5^ CFU/mL (Figure 4). After 20 h of incubation at 37 °C, the microbicide activity was evaluated using the tenfold serial microdilution technique. The serial dilutions and plate counts were performed in duplicate, and after inoculation and incubation at 37 °C for 24 h, the microbial colonies were counted and converted to colony-forming units per milliliter (CFU/mL). Each test was performed in triplicate and repeated on at least three separate occasions.

The quantitative evaluation of the antimicrobial effects was determined based on the viable cell counts of the microbial cells grown in liquid medium, quantified 20 h after adding the sample specimens.

### 2.11. MTT Assay 

L929 fibroblasts (5 × 10^5^ cells/well) were cultured in DMEM (Dulbecco’s Modified Eagle Medium, Sigma-Aldrich) [120] medium supplemented with 10% fetal bovine serum (Sigma-Aldrich) and 1% pen/strep (penicillin/streptomycin solution, Sigma Aldrich) for 24 h at 37 °C in 95% humidity with 5% CO_2_. 

The samples were co-cultured with the fibroblasts for 24 h (37 °C, 95% humidity, 5% CO_2_). The MTT (3-(4,5-dimethylthiazol-2-yl)-2,5-diphenyltetrazolium bromide) [121] assay was used to evaluate the cell viability and proliferation in the presence of the materials. Cells were incubated for 4 h with MTT reagent (Vybrant^®^ MTT Cell Proliferation Assay Kit, V-13154) at 37 °C in 95% humidity with 5% CO_2_. After incubation, formazan crystals were solubilized with DMSO for 10 min at room temperature. The absorbance was measured at λ = 540 nm using the Multiskan FC apparatus (Life Technologies Holdings Pte. Ltd., a part of Thermo Fisher Scientific Inc., Singapore). The cell morphology was evaluated using an inverted Olympus IX73 microscope after 24 h of incubation with the biomaterials.

### 2.12. Statistical Analyses

The antimicrobial and MTT tests results were analyzed using one-way ANOVA repeated measurement test. All statistical analyses were performed using GraphPad Prism Software, version 5.03 428 (GraphPad Software, La Jolla, CA, USA). A significant difference was noted as * for *p* < 0.05, ** for *p* < 0.01, and *** for *p* > 0.001. 

## 3. Results and Discussions

### 3.1. Contact Angle

When using bioactive dressings to treat bedsores or pressure ulcers, an important property to consider is their hydrophilicity. When the polymeric matrix in the dressing comes into contact with the wet parts of the skin lesion, due to its hydrophilicity, the adsorption of the liquid components exuded from the dermis on the polymer takes place, and then the progressive absorption of biological fluids from the wound by the polymer mass occurs. At the same time, the polymeric matrix swells and yields the active substances that play a binding role. In other words, an increasingly intense exchange takes place. An important method of analysis used to prove the hydrophilicity of the polymeric matrices is the analysis of the contact angle and the free surface energy. 

All solid components of polymeric matrices, namely polyvinyl alcohol, polyvinyl pyrrolidone, and carboxymethyl cellulose, have polar groups which determine their hydrophilic character and solubility in water. Plasticizer polyethylene glycol possess polar oxygen atoms and non-polar groups (CH)_2_ that allow it to be used as a carrier of many hydrophilic and hydrophobic active substances with the ability to enhance the material’s aqueous solubility or dissolution characteristics [82,122,123]. Glycerol, the second plasticizer used in the polymeric matrices, has three atoms of carbon in its molecules, each being bound to hydroxyl groups, which make it highly soluble in water. Determining the contact angle of polymeric matrices is important because it provides information on the degree of hydrophilicity and, consequently, the rate of absorption of the aqueous liquid medium. The compositions of the polymeric matrices described above enable a faster wettability and a significant decrease in the contact angle to be obtained during the measurements that used water. The results were statistically processed, determining the mean, median, and inter-quartile interval. The statistical parameters are summarized in the boxplot diagrams presented in Figure 5.

The values of the contact angles of the compositions of the capsules loaded with essential oils (E3—23.31 ± 0.49°, E5—20.13 ± 0.46°, E7—32.08 ± 1.19°) were higher compared to the corresponding compositions (E2—15.65 ± 0.66°, E4—18.62 ± 0.25°, E6—15.81 ± 0.98°, E8—12.32 ± 0.23°) in which the essential oils were mixed directly into the polymer matrix. Additionally, compositions E3, E5, and E7, including capsules containing essential oils showed higher contact angle values than the control sample E1. In contrast, compositions E2, E4, E6, and E8, in which the essential oils were uniformly dispersed in the polymer matrix, showed lower contact angle values than the control sample E1. This provides information indicating that the essential oils mixed directly with the components of the polymeric receptors caused an increase in the hydrophilicity and, consequently, a decrease in the swelling time of the materials in contact with biological fluids. Compositions E3, E5, and E7 showed a longer time of water penetration and swelling of the matrix and capsules, which resulted in a longer release of the essential oils. The results were compared using one-way ANOVA with a significance level of α = 0.05. The test was performed for all the data sets (results from E1–E9) and groups (E1, E3, E5, E7, E9 and E1, E2, E4, E6, E8), and all the test results revealed that not all the means were equal, and the treatment (change in composition) did not have an influence on the response (contact angle value). 

In comparison, other researchers found that the composition of PVA/PVP (50:50) exhibits a contact angle of 61.3 ± 3.1°, which decreases in value to 38.1 ± 2.1° with a higher content of AgNPs. This increased value of the contact angle is due to the 50% wt. content of the PVP [124]. In another study, the matrices of PVA/PVP (50:50) were found to exhibit a contact angle value of 52.3 ± 1.2° that increased to 79.6 ± 1.7° with a higher content of CuO-VB1 NPs [125].

### 3.2. Optical Microscopy

The various types and contents of the essential oils and polymeric components have influences on the shape and appearance of microcapsules, as can be observed visually.

As we can see in the optical microscope images (Figure 6), the microcapsules loaded with pine essential oil present with a flexible ovoid shape. The surfaces have a smooth, glossy appearance, without defects or cracks. The microcapsules loaded with thyme essential oil also have a flexible ovoid shape, and the surfaces have an opaque appearance, without defects or cracks. The microcapsules loaded with peppermint essential oil have a predominantly flexible spherical shape and a glossy, smooth surface, without cracks or defects. The capsule size and morphology were investigated using images obtained by the optical microscope.

The dimensions of the capsules loaded with essential oils are presented in the Table 5.

The morphology of the samples based on PVA and PVP was analyzed by optical microscopy (Figure 7) to highlight the combining compatibility of the components. The E1 control sample shows a uniform morphology, with a clear, transparent, smooth surface, without any distinction of the component materials. The E2 sample, loaded with fennel essential oil, has a smooth, slightly opaque surface. There are a few agglomeration points of the essential oil evenly distributed in the polymer matrix. Sample E3 contains microcapsules loaded with peppermint essential oil, which is in direct contact with the polymer matrix. The microcapsule shown in the image retains its shape and dimensions, and there are no gaps or cracks between the microcapsule and the polymer matrix. The E4 sample, loaded with peppermint essential oil, has a slightly opaque appearance, and the phase of the essential oil, which is evenly distributed in the polymer matrix, is easily differentiated. Sample E5 contains microcapsules loaded with pine essential oil, which is effectively fixed in the polymer matrix and retains its shape and dimensions. Figure 7 shows the microemulsion of the pine essential oil in the microcapsule. Sample E6, containing pine essential oil, is characterized by a uniform morphology, in which a small number of points of accumulation of the pine essential oil can be observed. Sample E7 contains microcapsules loaded with thyme essential oil. Figure 7 shows the occurrence of the microemulsion in the microcapsule and reveals the direct contact between the microcapsule and the polymeric matrix, without gaps or cracks. The E8 sample, loaded with thyme essential oil, has a uniform morphology, without any differentiation of the component phases and without gaps or other defects. Optical microscopy is a very effective method for studying morphology of polymeric compositions in order to highlight the morphological characteristics of the material and the changes that occur during physical or biological experiments. For this reason, numerous studies carried out on polymer matrices composed of the PVA/PVP tandem for medical applications were performed by optical microscopy [126,127].

### 3.3. FTIR Analyses

FTIR analyses were used to determine the functional groups that correspond to the loading of the essential oils into polymeric matrices. Figure 8 illustrates the spectra from the FTIR spectroscopy of the initial control samples E0_1 and E0_2. The E0_1 spectra present with bending and stretching vibration bands of the functional groups generated during the blending of the PVA and PVP in the aqueous solution. The E0_2 spectra present the changes in the bands that occurred after crosslinking with the glutaraldehyde of initial matrix PVA/PVP.

The main bands corresponding to vibrations in the IR domain of the initial compositions are presented in Table 6.

The FTIR spectra of E0_1 and E0_2 show a broad band with the peak between 3250 cm^−1^ and 3275 cm^−1^, corresponding to stretching vibrations of the OH hydroxyl groups, which formed intermolecular hydrogen bonds [128], and the peaks at around 2925 cm^−1^ can be assigned to asymmetric stretching vibrations of the -CH_2_ methylene groups from the PVA [129]. Peaks in the range of 1745–1680 cm^−1^ are assigned to symmetric stretching vibrations of the C=O groups [130], peaks at 1085 cm^−1^ are assigned to C-N stretching vibrations [131], and bending vibrations of CH from PVP are revealed at around 1428 cm^−1^, 1285 cm^−1^, and 920 cm^−1^. The changes that occurred in the PVA/PVP matrices during crosslinking were highlighted in the FTIR spectra by peaks of 1234.22 cm^−1^, assigned to the secondary stretching of C=O from GTA, which was present only in E0_2.

Figure 9 presents the FTIR spectra of the control sample E1 and compositions loaded with essential oils in the polymeric matrices. The FTIR spectra of control sample E1 show similarities with E0_2 due to the high content of PVA/PVP in the compositions. The same peaks can be observed with very small differences in the wavenumber due to the additional components. In addition, a peak at 811.885 cm^−1^ appears in E1 sample, corresponding to the content of plasticizers, stabilizers, and antioxidants. The FTIR spectra of the polymeric matrices loaded with essential oils are similar to those of the control sample E1 (Figure 9). Peaks in the domain ranging from 4000 cm^−1^ to 600 cm^−1^ are maintained, some of them with very small differences in the wavenumber. In contrast, a peak of 811.885 cm^−1^, observed in E1, is not present in the other spectra of the samples E2–E8. The presence of essential oils in the matrices is related to peaks at very low wavenumbers, such as the domain between 926 cm^−1^ and 800 cm^−1^ for the fennel, pine, thyme, essential oils, assigned to deformation vibrations of C-H in the aromatic cycle, and between 628 cm^−1^–603 cm^−1^ for the peppermint essential oil, assigned to deformation vibrations of S– or Cl–C in the aromatic cycle [132].

Figure 10 presents the FTIR spectra of the samples containing microcapsules loaded with essential oils. The FTIR spectra of the microcapsules loaded with essential oils are similar to that of the control sample E1. 

### 3.4. Gel Fraction

The value of the gel fraction content in the E0_2 compositions was determined to be 56.64%. This means that around 43% of the content of the E0_2 compositions is soluble in the aqueous liquid surrounding media. 

The results obtained for the swelling degree (SD), water solubility (WS), water vapor transmission (WVTR), and water permeability (WVP) are presented in Table 7. It is well known that the swelling capacity and water solubility are network parameters that can affect the active principle’s release. At the same time, the water absorption capacity indicates the ability of the hydrogel matrix to absorb and retain wound secretions, promoting a proper oxygen supply [133,134]. Therefore, swelling is vital for creating a good wound dressing material [135]. The obtained values for the water uptake at equilibrium (reported as SD), showing a high degree of swelling, are acceptable for the purposes of this study and can be correlated with the film composition. The highest values were obtained for E2 and especially for E8. Similar findings were previously reported for dialdehyde cellulose crosslinked poly(vinyl alcohol) hydrogels [136] and for other transdermal drug delivery systems [137,138].

The value obtained for the water solubility of the E0_2 compositions is the smallest of all the compositions analyzed and reflects the characteristics of sample, composed of only two materials (PVA and PVP) that are soluble in water, with approx. 40% wt. of dried substance in deionized water. In contrast, the other compositions have higher values of solubility in water, as they contain dry mass and liquid components: plasticizers, antioxidants, and essential oils, which diffuse easily in aqueous medium. The results were analyzed using one-way ANOVA. It was found that the composition of the samples significantly influenced the WS (*p* = 0.033).

Water solubility, as a characteristic of biodegradable films, creates potential benefits [137]. Ideally, the wound healing rate should be in accordance with the degradation rate of the polymeric matrix [138]. It can be seen from Table 7 that the WS depends on the film composition. The highest values were obtained for the E4 and E5 films. All films kept their original shape during the experiments of the water solubility [139].

A good wound dressing must allow for exudate evaporation in order to facilitate the healing process and, at the same time, it should prevent wound dehydration [138]. A high WVTR induces dehydration and dressing adhesion to the wound surface, while a WVTR that is too low leads to exudate accumulation, enabling microbial growth and wound infection [137]. Considering the generally accepted WVTR values in the range of 76–9360 gm^−2^ day^−1^, depending on the skin condition and wound type, and the specific range of 904–1447 gm^−2^ day^−1^, characterizing wounds with moderate exudates, we can conclude that our films could be applied to wounds with low to moderate exudates [138]. Other researchers have determined the WVTR of PVA in different compositions combined together with chitosan or collagen, and the values obtained were higher than those of our compositions [140].

Low WVP values are known to provide a good healing environment by maintaining the optimal moisture content of the dressing materials. Our obtained values for the WVP are lower, for instance, than those reported for carrageenan-based functional hydrogel films [134].

### 3.5. Scanning Electron Microscopy and Energy Dispersive X-ray Analysis

The morphologies and the compositions of the polymeric matrices E1–E8 are presented in Figure 11, together with the EDAX spectra. The SEM micrographs, for the bulk of samples, were obtained in order to characterize the microstructure. The control sample E1 exhibits a homogeneous porous morphology, and no phases can be distinguished in the material. The components are compatible and well mixed. The micrograph of the E2 sample shows a uniform morphology of the polymeric matrix, with some traces of fennel essential oil, but generally phases are well mixed. Sample E3 shows a uniform morphology of the material and some microcapsules loaded with peppermint essential oil. The microcapsules are well fixed in the bulk material, and no empty spaces around them were observed. The micrograph of sample E4 shows a uniform porous morphology, and peppermint essential oil is spread homogeneously over the matrix. Sample E5’s micrograph shows a uniform morphology of the matrix, and the microcapsules loaded with pine essential oil are well fixed in the material. Sample E6’s micrograph shows a uniform morphology of the matrix, but some traces of pine essential oil can be observed. Sample E7’s micrograph shows a uniform homogeneous morphology of the polymeric matrix, with some microcapsules filled with thyme essential oil that are well fixed in the bulk material, without cracks or holes. Sample E8’s micrograph shows some traces of thyme essential oil in the polymeric matrix. 

The EDAX spectra of samples E1, E2, E4, E6, and E8 reveal two main components of the matrices, namely carbon and oxygen. In contrast, the EDAX spectra of samples E3, E5, and E7 show the two major components of the polymeric matrix to be carbon and oxygen, together with another two main components covering the shell of the microcapsules, namely Ca and Cl.

### 3.6. Antimicrobial Assays

The antimicrobial efficiency was evaluated for all the samples, E1–E8. For the qualitative screening of the antimicrobial effects, the results were evaluated by observing the growth inhibition zone obtained after placing square specimens of the samples on solid medium streaked with microbial inoculum. After 20 h of contact, the inhibition zones were observed and measured. Figure 12 presents a graphical representation of the inhibition zone diameters yielded by the tested microbial strains after 20 h of contact with the samples. In some cases, the inhibition zones extended beyond the direct contact area between the sample and the solid surface of the medium. The values of the diameters of these areas were measured and included.

The qualitative tests showed that both the samples loaded with thyme essential oil (E7 and E8) expressed the largest clear inhibition zone with respect to all the microbial strains. Additionally, it can be observed that the antimicrobial efficiency of the sample loaded with thyme EO microcapsules was higher than the efficiency of the sample loaded with the thyme EO mixture. The only exception was the *Pseudomonas aeruginosa* ATCC 27853 Gram-negative strains, which showed the weakest sensitivity to the actions of all tested samples.

Additionally, the samples loaded with peppermint essential oil expressed a good inhibitory effect on the Gram-positive strains but not the Gram-negative strains, which are more selective. A potentiation of the inhibitory effect was also observed in this case when the peppermint essential oil was incorporated into the microcapsules. Sample E5, containing capsules loaded with pine essential oil, also expressed a good inhibitory effect on the Gram-positive strains but not the Gram-negative strains, in comparison with the control sample E1. Sample E6, loaded in the mixture with pine essential oil, expressed a low inhibitory effect on the Gram-positive and Gram-negative strains.

Compared with the growth of the first control sample (microbial strain culture in normal conditions) and second control sample (microbial strain culture in the presence of the unloaded sample E1), all the tested materials showed inhibitory effects, with a drastic attenuation of the corresponding CFU/mL logarithmic values (Figure 13) of the samples loaded with thyme essential oil (E7 and E8). An exception was observed in the case of the *Pseudomonas aeruginosa* ATCC 27853 Gram-negative strain, confirming the qualitative results indicating the high resistance of this strain to the essential oil’s inhibitory activity.

The quantitative results also confirmed our observation regarding the good inhibitory effect expressed by the samples loaded with peppermint essential oil on the Gram-positive strains and yeast strain *Candida albicans* ATCC 10231.

### 3.7. MTT Assay

The cell morphologies, evaluated using samples of the E1–E8 compositions, are presented in Figure 14. The cell viability and proliferation are presented in Figure 15.

Among the tested samples, E5 led to the highest level of cell proliferation, as revealed by the MTT test (Figure 14). Compared to both the unstimulated cells and the E1 control, samples E5, E7, and E8 stimulated the proliferation of L929 fibroblasts. Samples E6 and E2 were similar to the E1 control in terms of their cell viability and proliferation. Samples E3 and E4 induced the lowest level of proliferation of L929 fibroblasts. However, these samples were not toxic and did not lead to cell death, as shown by the phase contrast microscopy analysis (Figure 15). Importantly, none of the samples evaluated induced significant changes in the cell morphology.

Thus, the characterization of the physicochemical properties of compositions E1–E8 demonstrated that the constituents and their percentages were selected appropriately, the preparation stages were correctly carried out, and the sequence of the operations was normal and optimal. The biological analyses performed on the polymer compositions prepared by us in the laboratory allowed us to represent their antimicrobial and biocompatibility properties with certainty.

## 4. Conclusions

Currently, the topical treatment of bedsores and pressure ulcers is based on dressings that may contain antibiotics and chemically synthesized antimicrobial active substances. Given the healing properties of essential oils, we created a selection of polymeric matrices based on PVA/PVP, into which we introduced four types of essential oils (fennel, peppermint, pine, and thyme EO) by mixing them with the components of the polymer matrix or encapsulating and dispersing them in a bulk material. 

The morphologies of the samples, determined by optical microscopy, were different from each other. Samples E2, E4, E6, and E8, in which the essential oils were distributed in the polymer mass, showed a uniform morphology of the matrix, but traces of essential oils were highlighted. In contrast, samples E3, E5, and E7 showed only a uniform morphology of the polymer matrix, without phase distinction, and only in the case of the microcapsules loaded with essential oils. The presence of essential oils was demonstrated by FTIR analyses for compositions E2–E8. The SEM-EDAX analyses showed a uniform morphology of all the samples, with some differences, namely in the case of the E2, E4, E6, and E8 compositions, which showed traces of essential oils in the polymer mass as well as the pores. In contrast, in the E3, E5, and E7 compositions, into which the essential oils were loaded after encapsulation, the morphology was uniform, homogeneous, and without pores, as in the case of the control sample E1.

The determination of the contact angle and barrier properties of all the compositions showed that they exhibit hydrophilicity and can be used in contact with skin lesions in stages 1–3.

The biological analyses (antimicrobial and MTT assays) demonstrated, firstly, the diffusion of the essential oils from both types of compositions. The essential oils showed a good and selective inhibitory effect on, or antimicrobial properties against, the tested microorganisms of *Staphylococcus aureus ATCC 25923, Enterococcus faecalis ATCC 29212, Escherichia coli ATCC 25922, Pseudomonas aeruginosa ATCC 27853,* and *Candida albicans ATCC 10231.* According to the MTT assay, the compositions loaded with essential oils were not toxic and did not lead to cell death. None of the materials tested induced significant changes in the cell morphology of the fibroblasts when tested using the sample materials loaded with essential oils.

Commercialized wound dressings offer limited possibilities of use, because their main role is to absorb exudates from open wounds, having a limited antimicrobial effect. According to the results obtained from our physicochemical and biological analyses, we conclude that we obtained improved polymer compositions based on polyvinyl alcohol/polyvinyl pyrrolidone (PVA/PVP), because they possess important properties for healing open wounds (bedsores, pressure ulcers, etc.). An important aspect for the healing process is that the absorption of exudates from the wound takes place simultaneously with the release of vital components (vitamins, active substances, essential oils). We used essential oils obtained from mint, pine, fennel, and thyme, which ensured the destruction of a wide range of microorganisms (*Staphylococcus aureus ATCC 25923, Enterococcus faecalis ATCC 29212, Escherichia coli ATCC 25922, Pseudomonas aeruginosa ATCC 27853, Candida albicans ATCC 10231*). Currently, the HOFIGAL company produces lotions and ointments containing essential oils, which we used in our study for the treatment of various skin diseases. The use of essential oils in wound dressings offers positive evolutionary perspectives on the efficiency of the process of the treatment of acute skin wounds. Future recommendations and new research directions will examine more diverse types of essential oils used in dressings and perform biological evaluations and clinical testing.

## Figures and Tables

**Figure 1 materials-15-06923-f001:**
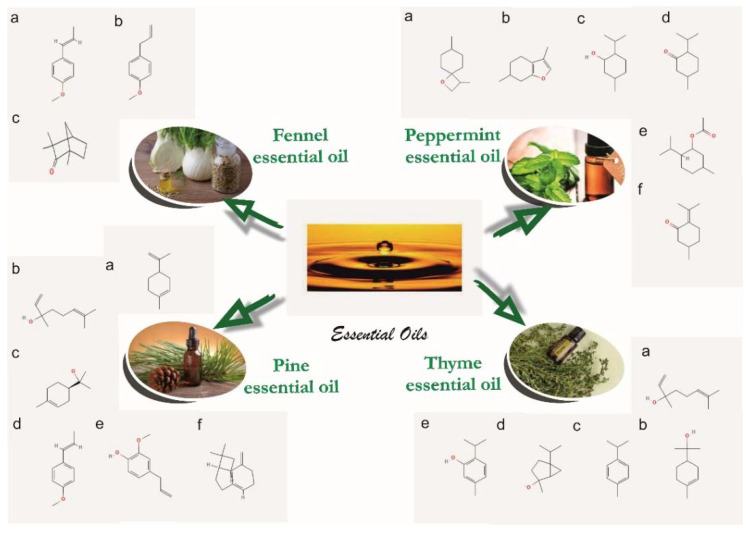
The composition of the essential oils: fennel essential oil [96,97] (IUPAC name: (a) 1-methoxy-4-[(E)-prop-1-enyl]benzene; (b) 1-methoxy-4-prop-2-enylbenzene; (c) 1,3,3-trimethylbicyclo [2.2.1] heptan-2-one); peppermint essential oil [98,99] (IUPAC name: (a) 3,7-dimethyl-1-oxaspiro[3.5]nonane; (b) 3,6-dimethyl-4,5,6,7-tetrahydro-1-benzofuran; (c) 5-methyl-2-propan-2-ylcyclohexan-1-ol; (d) 5-methyl-2-propan-2-ylcyclohexan-1-one; (e) (5-methyl-2-propan-2-ylcyclohexyl) acetate; (f) 5-methyl-2-propan-2-ylidenecyclohexan-1-one); pine essential oil [100,101] (IUPAC name: (a) 1-methyl-4-prop-1-en-2-ylcyclohexene; (b) 3,7-dimethylocta-1,6-dien-3-ol; (c) 2-[(1R)-4-methylcyclohex-3-en-1-yl]propan-2-ol; (d) 1-methoxy-4-[(*E*)-prop-1-enyl]benzene; (e) 2-methoxy-4-prop-2-enylphenol; (f) *trans*-(1*R*,9*S*)-8-methylene-4,11,11-trimethylbicyclo [7.2.0]undec-4-ene); thyme essential oil hydrate [102,103] (IUPAC name: (a) 3,7-dimethylocta-1,6-dien-3-ol; (b) 2-(4-methylcyclohex-3-en-1-yl)propan-2-ol; (c) 1-methyl-4-propan-2-ylbenzene; (d) 2-methyl-5-propan-2-ylbicyclo[3.1.0]hexan-2-ol; (e) 5-methyl-2-propan-2-ylphenol).

**Figure 2 materials-15-06923-f002:**
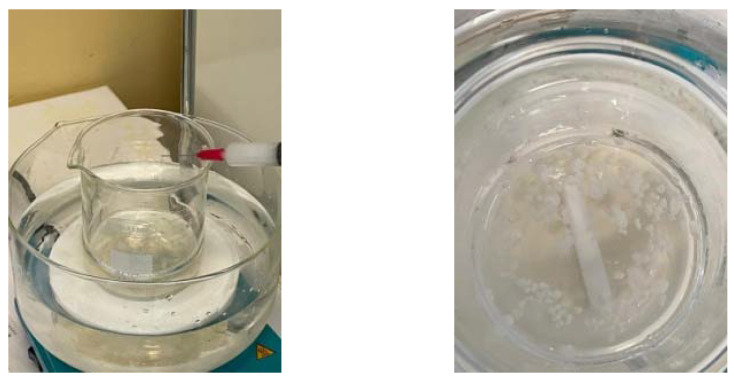
Performing the microencapsulation of essential oils in a sodium alginate shell.

**Figure 3 materials-15-06923-f003:**
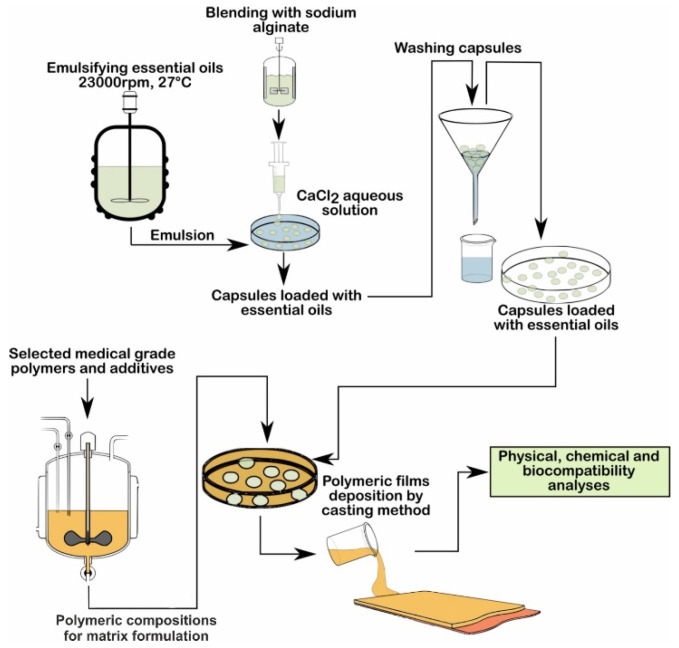
Flow chart of the preparation of samples E1–E8.

**Figure 4 materials-15-06923-f004:**
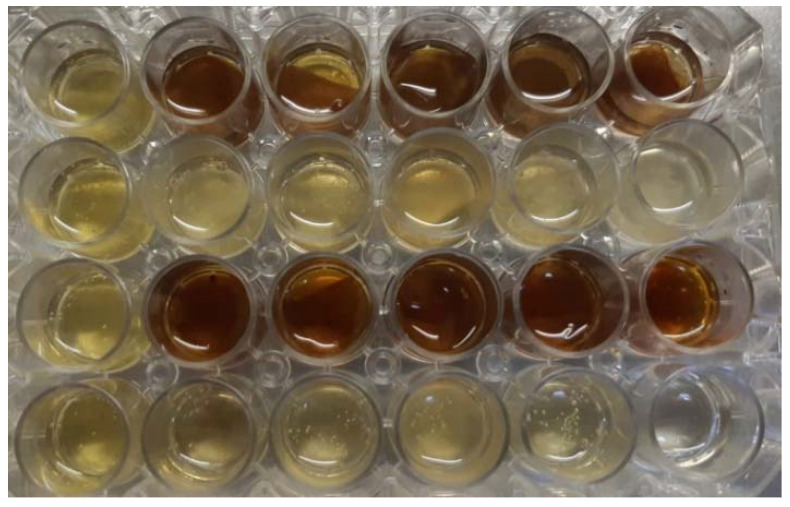
Well plates with samples E1–E8.

**Figure 5 materials-15-06923-f005:**
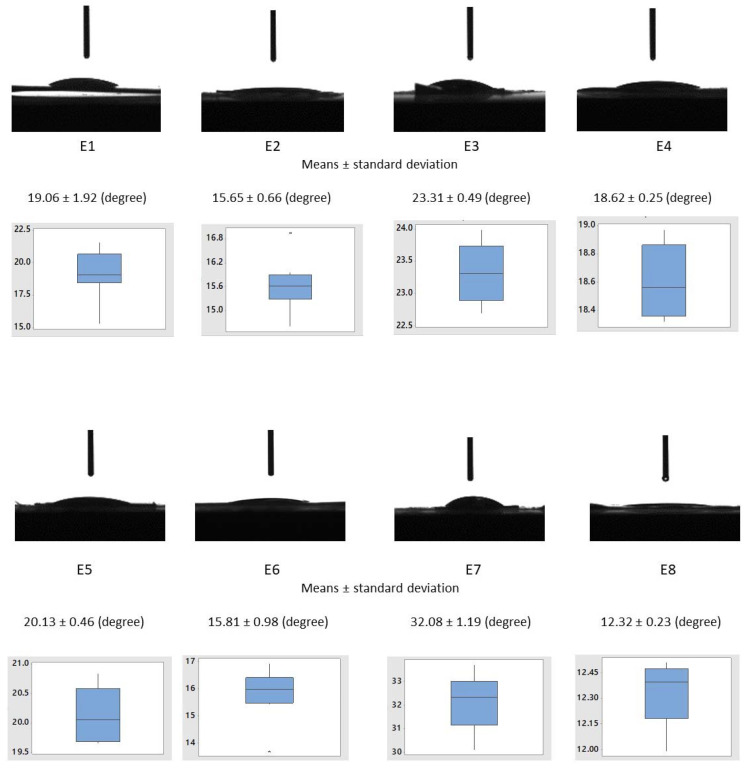
Water contact angles on the polymeric matrices’ surfaces: E1–E8.

**Figure 6 materials-15-06923-f006:**
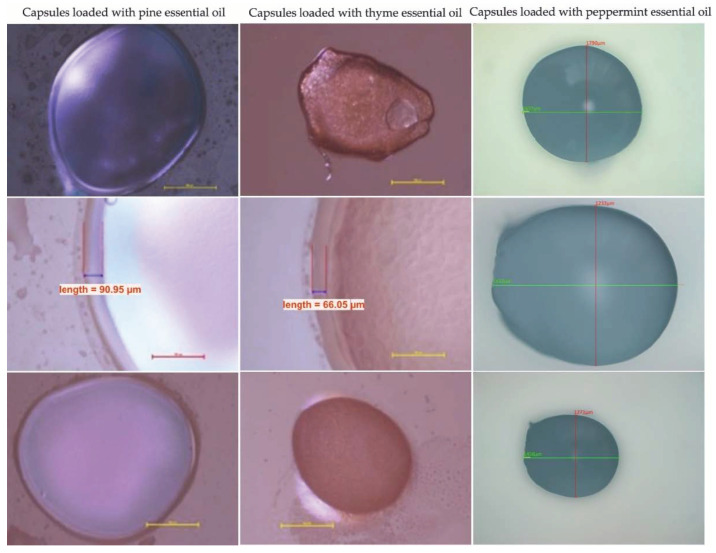
Examples of optical microscopy images of capsules loaded with pine, thyme, and peppermint essential oils.

**Figure 7 materials-15-06923-f007:**
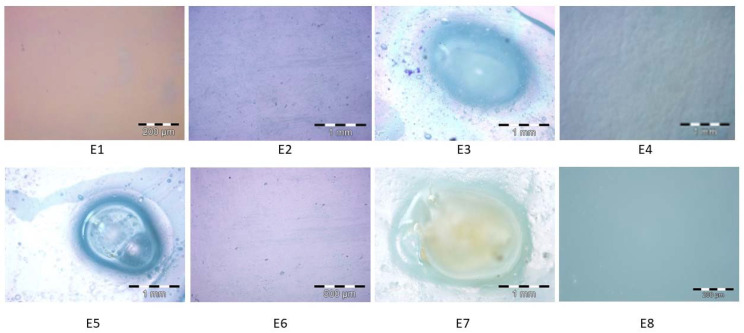
Examples of optical microscopy images of polymeric compositions loaded with essential oils extracted from fennel, peppermint, pine, and thyme (E1–E8).

**Figure 8 materials-15-06923-f008:**
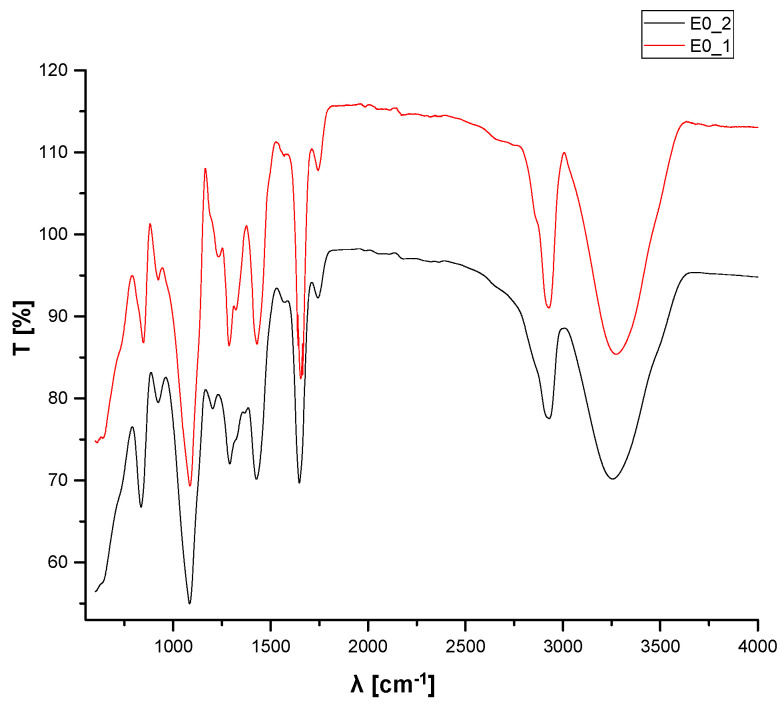
Spectra from the FTIR spectroscopy of the E0_1 and E0_2 compositions.

**Figure 9 materials-15-06923-f009:**
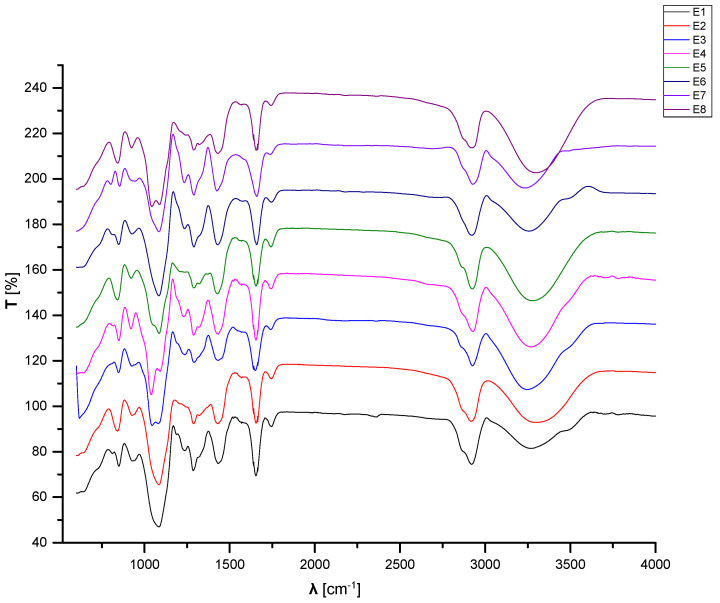
FTIR spectroscopy spectra of the E1–E8 samples.

**Figure 10 materials-15-06923-f010:**
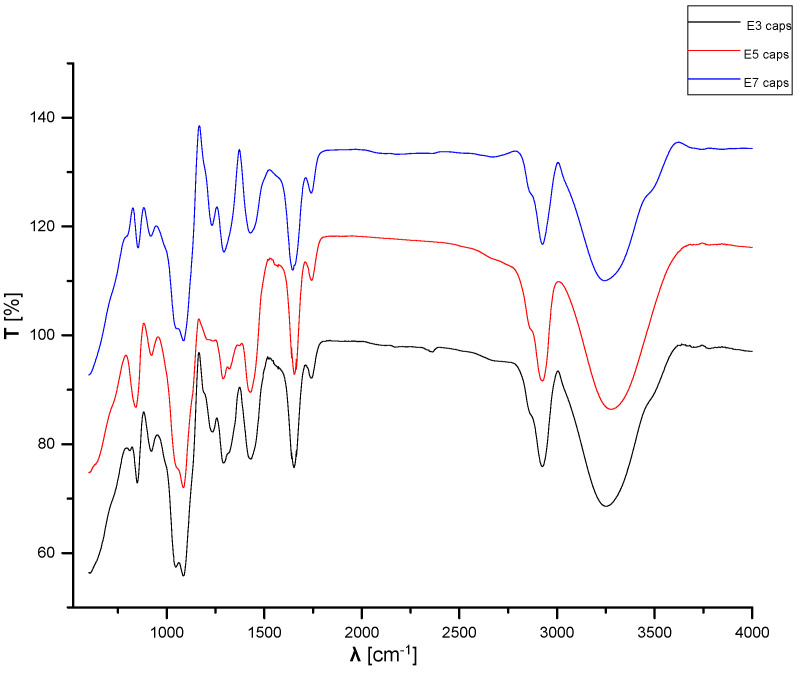
FTIR spectroscopy spectra of samples E3, E5, and E7.

**Figure 11 materials-15-06923-f011:**
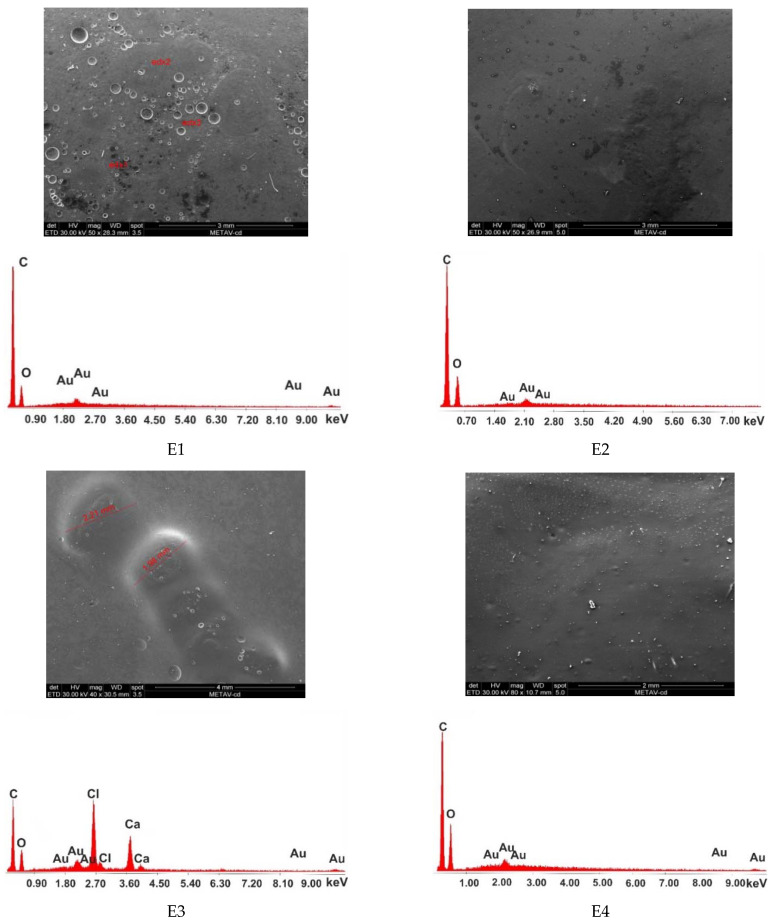
SEM micrographs and EDAX diagrams of compositions E1–E8.

**Figure 12 materials-15-06923-f012:**
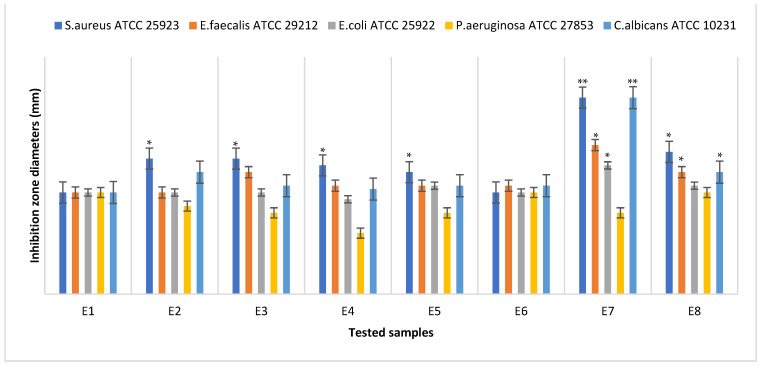
Graphical representation of the inhibition zone diameters yielded by the tested microbial strains after 20 h of contact with the samples E1–E8. Significant evidence of an inhibitory effect, expressed as the inhibition zone diameters, was noted as * for *p* < 0.05 and ** for *p* < 0.01.

**Figure 13 materials-15-06923-f013:**
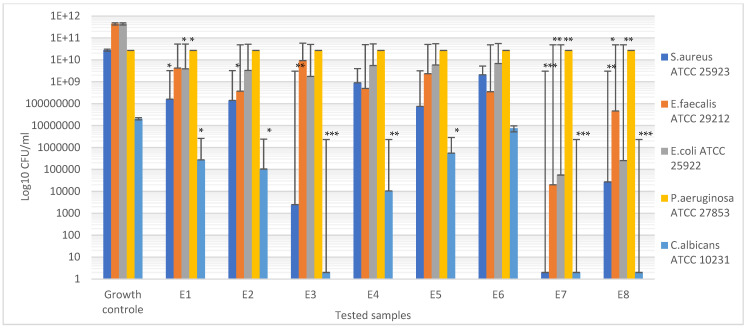
Graphical representation of the CFU/mL values evaluating the inhibitory effect. Significant evidence of the inhibitory effect on the bacterial growth manifested by the tested samples was noted as * for *p* < 0.05, ** for *p* < 0.01, and *** for *p* > 0.001.

**Figure 14 materials-15-06923-f014:**
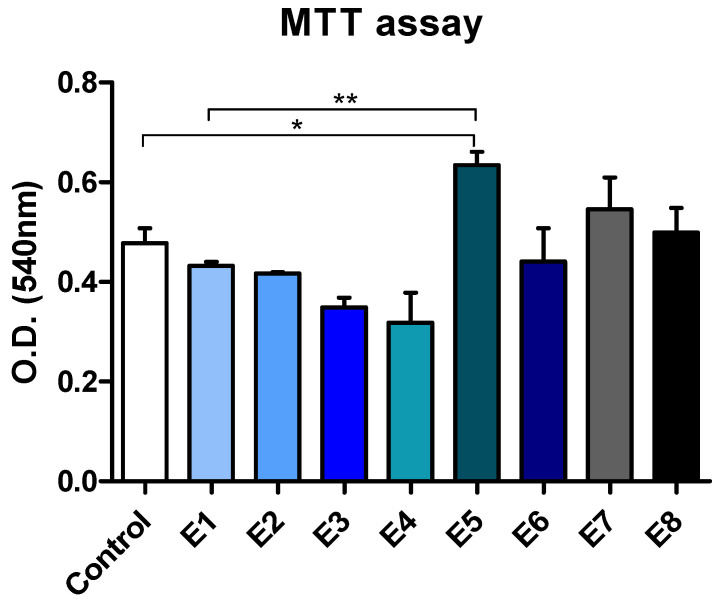
L929 cell viability and proliferation measured by the MTT test. O.D., optical density (statistical significance: * *p* < 0.05, ** *p* < 0.01). The control is represented by unstimulated cells in standard culture conditions.

**Figure 15 materials-15-06923-f015:**
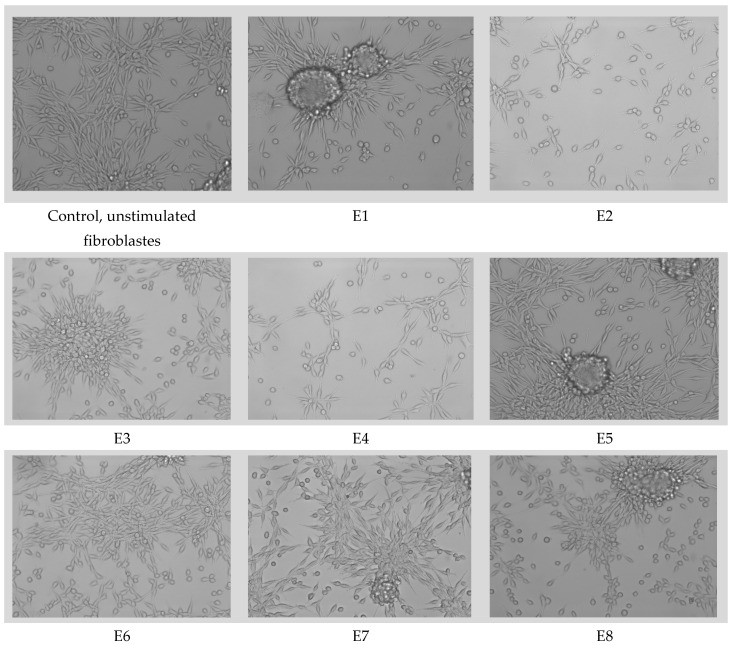
Cell morphology of the fibroblasts for the sample materials loaded with essential oils.

**Table 1 materials-15-06923-t001:** Essential oils: properties and applications [28,48].

Essential Oil	Property	Application
Eucalyptus (*Eucalyptus globulus*)	Antibacterial, antispasmodic, and antiviral	Effective in combating *Staphylococccus aureus* infections.
Frankincense (*Boswellia carteri, Frereana & Sacra*)	Anti-inflammatory	*Boswellic acid* can improve the immunesystem. Beneficial effects in topical applications used to heal pain and inflammation.
Lavender *(Lavandula angustifolia)*	Calming properties	Accelerating the healing time for burns, cuts, stings, and other wounds; decreasing oxidative stress.
Peppermint (*Mentha piperita*)	Antibacterial	Antibiotic-resistant.
Thyme (*Thymus* sp.)	Antibacterial	Treatment of cutaneous lesions, acts against *Staphylococcus aureus* and *Klebsiella pneumoniae*.
Pine (*Pinus sylvestris*)	Antibacterial	Treatment of infections with *Staphylococcus aureus*, *Escherichia coli*, and *Candida albicans.*
Fennel (*Foeniculum vulgare*)	Antibacterial	Treatment of infections with *Salmonella enteritidis* and *Salmonella typhimurium*.

**Table 2 materials-15-06923-t002:** Physical-chemical properties of fennel, peppermint, pine, and thyme essential oils.

Property	Fennel Essential Oil	Peppermint Essential Oil	Pine Essential Oil	Thyme Essential Oil
Relative density [g/cm^3^]	0.961–0.975	0.900–0.916	0.855–0.875	0.915–0.935
Refraction index	1.528–1.539	1.457–1.467	1.465–1.480	1.490–1.505
Optical rotation, [°]	10.0–24.0	−30–−10	−9.0–30.0	−21.0–+15.0
Residue on evaporation, [%]	max 1.5	max 1.5	max 1.5	max 1.5
Solubility in:	hexane	hexane	hexane	hexane
Antioxidant activity (mg equivalent to Fe_2_SO_4_ × 7H_2_O/g for sample)	6.09	6.07	7.09	6.79

**Table 3 materials-15-06923-t003:** Composition of fennel, peppermint, pine, and thyme essential oils determined by gas chromatography coupled with a mass spectrometer (GC-MS).

Peppermint Essential Oil		Thyme Essential Oil		Pine Essential Oil		Fennel Essential Oil	
Comp.	RT	Comp.	RT	Comp.	RT	Comp.	RT
β-Pinene	7.035	α-Pinene	5.21	α-Pinene	4.83	α-Pinene	7.02
Sabinene	7.909	α-Phellandrene	5.42	Camphene	6.16	Camphene	8.75
d-Limonene	12.657	α-Myrcene	12.84	β-Pinene	7.94	β-Pinene	10.68
Eucalyptol	12.922	4-Carene	13.23	3-Carene	10.63	β-Phellandrene	11.55
o-Cymene	16.462	Cineole	14.42	β-Myrcene	12.04	γ-Phellandrene	13.96
Isomenthone	23.829	3-Carene	16.28	d-Limonene	13.40	Myrcene	14.28
d-Menthone	24.754	o-Cimene	17.26	β-Phellandrene	13.74	d-Limonene	15.94
Menthol, acetate	27.353	Camphor	24.70	Terpinolene	16.98	Cineol	16.40
Isopulegol	27.533	Linalool	25.75	Cariofilene	26.22	α-Phellandrene	18.85
Caryophyllene	28.063	Caryophilene	26.90	Germacrene	28.94	γ-Terpinene	20.42
β-Terpineol	28.502	Terpinene-4-ol	27.04	α-Eudesmene	29.15	o-Cimene	21.12
Pulegone	29.614	Carvacol methyl ether	27.16	α-Murolene	29.33	Fenchone	28.26
Levomenthol	29.778	Bomeol	29.34	Elixene	29.49	Camphor	35.11
Terpineol	31.410	Isoledene	30.70	Cadinene	30.04	Estragol	44.73
Piperitone	31.930	Tymol	39.32	Cadinol	39.54	Anethole	53.57
		Carvacrol	39.84			Benzaldehyde	62.80

**Table 4 materials-15-06923-t004:** Composition of the experimental matrices E1–E8.

Sample/ Component	E1 [%]	E2 [%]	E3 [%]	E4 [%]	E5 [%]	E6 [%]	E7 [%]	E8 [%]
PVA	25	23	23	23		23	46.5	46.5
PVP	15	10	10	10	10	16.5	18	18
Sodium alginate	-	-				1 g in 100 mL water		
Tween 80						0.02		
CaCl_2_						30 g in 100 mL water		
CMC *	2	2	2	2	2	2	2	2
Fennel essential oil		12						
Peppermint essential oil			12	12				
Pine essential oil					12	12		
Thyme essential oil							12	12
Vitamin A	3	3	3	3	3	1.5	1.5	1.5
Vitamin E	2	2	2	2	2	2	2	2
Deionized water	50	50	50	50	50	14.5	14.5	14.5
Glycerol	2.5	2.5	2.5	2.5	2.5	3	3	3
Polyethylene glycol						8		
Glutaraldehyde	0.5	0.5	0.5	0.5	0.5	0.5	0.5	0.5
Zn stearate	0.6	0.6	0.6	0.6	0.6	0.6	0.6	0.6

* CMC—carboxymethylcellulose.

**Table 5 materials-15-06923-t005:** Characteristics of microcapsules loaded with pine, thyme, and peppermint essential oils.

Essential Oil Encapsulated	Wall Thickness [µm]	Diameter [µm]
Pine essential oil	89.40 ± 0.96	1523.31 ± 1.23
Thyme essential oil	66.05 ± 0.33	1142.49 ± 1.03
Peppermint essential oil	74.08 ± 0.89	1233.24 ± 0.74

**Table 6 materials-15-06923-t006:** Peaks assigned to specific vibrations of functional groups determined in the FTIR analysis of compositions E0_1 and E0_2.

E0_1		E0_2	
Wavenumber [cm^−1^]	Corresponding Band	Wavenumber [cm^−1^]	Corresponding Band
3256.22	Stretching OH from PVA	3273.57	Stretching OH from PVA
2926.45	Asymmetric stretching of CH_2_ from PVA	2925.48	Asymmetric stretching of CH_2_ from PVA
1740.44	Symmetric stretching of C=O from PVP	1742.37	Symmetric stretching of C=O from PVP
1428.03	Symmetric deformation of CH from PVA	1428.99	Symmetric deformation of CH from PVA
		1319.07	Stretching asymmetric due to CH-OH from ethanol
1289.18	Deformation of CH from PVA	1286.29	Deformation of CH from PVP
		1234.22	Stretching secondary C=O from GTA
1084.76	Stretching of C-N from PVP	1085.73	Stretching of C-N from PVP
921.807	Deformation C-H from PVP	923.736	Deformation C-H from PVP
835.026	Stretching C-C from PVA	845.633	Stretching C-C from PVA

**Table 7 materials-15-06923-t007:** Film thickness (m), swelling degree, water solubility (%), water vapor transmission rate (g m^−2^ day^−1^), and water vapor permeability of the composite films (g m^−1^ s^−1^ Pa^−1^).

Sample	δ × 10^3^, (m)	SD	WS, (%)	WVTR, (gm^−2^ day^−1^)	WVP × 10^10^, (g m^−1^ s^−1^ Pa^−1^)
**E0_2**	0.167 ± 0.003	7.019 ± 0.389	43.35 ± 0.73	624.9 ± 21.5	5.276 ± 0.034
**E1**	0.238 ± 0.003	4.847 ± 0.518	59.27 ± 0.68	578.5 ± 10.4	6.816 ± 0.123
**E2**	0.307 ± 0.030	6.472 ± 0.325	59.34 ± 0.61	493.3 ± 7.4	7.496 ± 0.112
**E3**	0.247 ± 0.003	5.775 ± 0.624	59.69 ± 0.97	575.9 ± 14.4	7.041 ± 0.176
**E4**	0.237 ± 0.003	6.048 ± 0.831	66.02 ± 1.05	521.2 ± 5.1	6.115 ± 0.596
**E5**	0.297 ± 0.004	4.654 ± 0.254	76.97 ± 1.03	604.9 ± 11.5	8.894 ± 0.170
**E6**	0.360 ± 0.044	3.735 ± 0.365	58.75 ± 0.94	612.9 ± 16.5	10.925 ± 0.296
**E7**	0.188 ± 0.012	3.206 ± 0.158	48.64 ± 0.65	606.8 ± 24.2	5.647 ± 0.225
**E8**	0.184 ± 0.027	9.631 ± 0.832	57.01 ± 1.32	134.7 ± 3.1	1.227 ± 0.028

## Data Availability

The experimental data on the results reported in this manuscript are available upon official request from corresponding authors.

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
