# Peer review of "Essential Oils as Antimicrobial Active Substances in Wound Dressings"

_materials, 2022, doi:10.3390/ma15196923_

Round 1
Reviewer 1 Report
Although there are interesting results in this work, the manuscript needs to be re-written. Firstly, the manuscript must be thoroughly checked for grammar, syntax and vocabulary mistakes. Secondly, I think that the R&D and conclusion sections are poorly written and downgrade the quality of the interesting results.
Here are a few suggestions:Abstract
1.The abstract has a very long introduction, is poorly written and does not include any of the outcomes of the present article. I believe the authors should focus on presenting the novelty of their research work as well as write the main outcomes of their research.
Materials and methods
2.The word “recipe” is inappropriate for a scientific article. The word “protocol” should be used instead. Furthermore, instead of the term “polymeric recipe”, I would suggest “polymeric matrix” as a more appropriate term throughout the manuscript.
Results and discussion
3.Line 357: Please replace “him” with “it”.
4.Sections 3.1-3.3 include no discussion. The authors simply present their data without explaining to the reader their outcomes. This is a combined R&D section, so I suggest a further discussion.
5. Generally, in this section I would suggest further discussion of the outcomes of this research. The authors move to the conclusion section without having discussed their findings.
Conclusions
6. The conclusion section does not attract a reader’s interest as it’s like summarizing the results. The authors should stress the importance of their findings in this section and not just highlight results that have already been mentioned in the R&D section.
Figures
7. Figures 1-4: the images that show the essential oils should be removed as they are from the internet. The authors should also number the chemical structures of molecules presented in the figure and link them with the names in the captions.
8. Figure 14. The text in EDAX spectra and axis is not clear. Authors should magnify these images in the figure.
9. Figure 23. The authors do not explain in the legend anything about the figure (e.g. the **, the DO- probably it’s optical density but not clear). Please revise.
Tables
10. Tables 2-3: Where did the authors obtain the composition? Is it published anywhere (e.g. company’s website or brochure)? The source of these data should be clarified.
11. Table 4: What were the concentrations of these substances?
Author Response
Dear reviewer,
The Authors of the manuscript entitled “Essential oils as antimicrobial active substances in wound dressings” submitted to MATERIALS thank the reviewer for reviewing our manuscript. We are deeply grateful for the observations and comments that we addressed and feel that greatly increased the quality of our manuscript. Please find below the answers to all comments and suggestions.

Reviewer 2 Report
The manuscript written scientifically well
Well design work
The technical aspect of work is good.
A request to include one heading of literature review and explain more in detail under that heading.
Future recomendations and directions need to explain and more detail need for future researchers.
Author Response

(The authors gave the same response as above.)

Reviewer 3 Report
Provide latin names for all plants used in this study.
It is not clear do you used commercial essential oil or laboratory obtained. Please provide more information.
Author Response

(The authors gave the same response as above.)

Reviewer 4 Report
Dear authors,
Firstly, I want to congratulate you on the valuable and significant study that has been presented. Then, I offer some remarks with respect and professionalism for the work performed to assist in the review process:
Abstract
The abstract is not properly written according to the journal's rules. For this session, the following are required: 1) Background: Place the question addressed in a broad context and highlight the purpose of the study; 2) Methods: Briefly describe the main methods or treatments applied. Include any relevant preregistration numbers, and species and strains of any animals used. 3) Results: Summarize the article's main findings; and 4) Conclusion: Indicate the main conclusions or interpretations. Please rewrite the abstract.
Line 27; line 176 and elsewhere in the text: be sure to state the meaning before using any abbreviations;
Materials and Methods
Why did the authors identify the molecular structure of the chemical constituents of essential oils only in Figure 3? Please proceed with the identification in figures 1, 2, and 4.
Table 4: Why did the authors not formulate microcapsules with fennel essential oil?
Lines 209 - 217: The text does not fit the objectives of the methodology and is much more similar to the discussion. Please adjust the methodology to make the text easier to understand.
Lines 221 and throughout the text: separate the unit from the numeral with a blank space.
Line 228: Where was the optical microscope manufactured? Please enter the required information. Additionally, how were the optical microscopy images made? What metric scale was used?
Results and discussions
Figure 8: Which unit is the contact angle measure expressed? Please insert it in the appropriate place.
Figure 8: The values represented on the y-axis are very difficult to read.
Lines 372-379: What statistical evidence supports the results presented? Demonstrate statistical differences are significant by the appropriate method. The differences shown in the angle measurements, as highlighted, are somewhat arbitrary.
Figure 9: The morphology and scale bar measurements are really difficult to read due to the small size.
Table 05: Was the data not read in triplicate? A measure of dispersion is much more interesting for a study of this nature.
The WVTR unit in table 7 is “g X m-2 X day-1” while in line 515 an analysis is presented in “g X m-2 X h-1”, thus the questioning follows: is it possible to establish this type of conclusion about the applicability of formulated films considering that the units differ?
Line 500: Are there statistically significant differences between E2 and the other treatments? Demonstrate using the appropriate statistical treatment to support your result.
Figure 15: again, a statistical test is required to support the results presented.
The data from figures 16-20 are repeated in figure 15 and this makes the text tiresome for the reader. Choose a single way to present them. Additionally, figure 15 does not show the y-axis. Include making the text easier to read.
Figure 21: again, a statistical test is required to support the results presented.
Figure 21: What does GRO... mean?
Finally, in general:
Most figures are grouped and the scales, axes, and morphological data are small, making the data difficult to read. Translating some important data from figures into tables, and inserting figures into supplementary material, can help with the logical reporting of the manuscript.
Statistical treatment is insufficient in the manuscript. The authors infer the meaning of the data collected very qualitatively, I advise you to subsidize the highlighted results through the appropriate statistical test.
In the microbiological analysis, positive control was not used, on the other hand, the E1 formulation (without essential oil or essential oil microsphere) showed antimicrobial potential. However, the magnitude of the antimicrobial potential is compromised because there is no adequate statistical treatment and the authors did not use a positive control. This represents a gap in the study. Considering these data, it is possible to predict that the essential oil is dispensable in the formulations. How do the authors intend to solve this problem?
Author Response

(The authors gave the same response as above.)

Round 2
Reviewer 3 Report
After the corrections, the manuscript is significantly improved. However, it is still unclear do you used commercial essential oils or you distilled them.
In first case it is necessary to quote producers, and in second case plant part and way in which you produced oils.
Author Response

(The authors gave the same response as above.)

Reviewer 4 Report
The manuscript has been sufficiently improved to warrant publication in Materials.
Author Response

(The authors gave the same response as above.)
